# Clinical improvement after surgery for degenerative cervical myelopathy; A comparison of Patient-Reported Outcome Measures during 12-month follow-up

Christer Mjåset[1,2,3]*, John-Anker Zwart[1,3], Frode Kolstad[2], Tore Solberg[4,5,6], Margreth Grotle[3,7]

1 Faculty of Medicine, University of Oslo, Oslo, Norway, 2 Department of Neurosurgery, Oslo University Hospital, Oslo, Norway, 3 Department of Research and Innovation, Division of Clinical Neuroscience, Oslo University Hospital, Oslo, Norway, 4 Institute of Clinical Medicine, The Arctic University of Norway, Tromsø, Norway, 5 Department of Neurosurgery, The University Hospital of North Norway, Tromsø, Norway, 6 The Norwegian Registry for Spine Surgery, The University Hospital of North Norway, Tromsø, Norway, 7 Department of Physiotherapy, Faculty of Health Sciences, Oslo Metropolitan University, Oslo, Norway

* chrmja@gmail.com

**Data Availability Statement:** The data cannot be shared publicly since it contains potentially identifiable personal information about the

## Abstract

### Object

Although many patients report clinical improvement after surgery due to degenerative cervical myelopathy, the aim of intervention is to stop progression of spinal cord dysfunction. We wanted to provide estimates and assess achievement rates of Minimal Clinically Important Difference (MCID) at 3- and 12-month follow-up for Neck Disability Index (NDI), Numeric Rating Scale for arm pain (NRS-AP) and neck pain (NRS-NP), Euro-Qol (EQ-5D-3L), and European Myelopathy Score (EMS).

### Methods

614 degenerative cervical myelopathy patients undergoing surgery responded to Patient-Reported Outcome Measures (PROMs) prior to, 3 and 12 months after surgery. External criterion was the Global Perceived Effect Scale (1–7), defining MCID as "slightly better", "much better" and "completely recovered". MCID estimates with highest sensitivity and specificity were calculated by Receiver Operating Curves for change and percentage change scores in the whole sample and in anterior and posterior procedural groups.

### Results

The NDI and NRS-NP percentage change scores were the most accurate PROMs with a MCID of 16%. The change score for NDI and percentage change scores for NDI, NRS-AP and NRS-NP were slightly higher in the anterior procedure group compared to the posterior procedure group, while remaining PROM estimates were similar across procedure type. The MCID achievement rates at 12-month follow-up ranged from 51% in EMS to 62% in NRS-NP.

participants. A record is kept on a secure server at Oslo University Hospital. Data access through the Norwegian Registry for Spine Surgery (NRSS) located in Tromsø, Norway, can be obtained for researchers who meet the criteria for access to confidential data. The registry can be contaced by email (nakkerygg@unn.no) or phone (+47 777 54287). Current managing director is Kjetil Samuelsen. More information can be found on the website (www.nakkeryggreg.no).

**Funding:** The author(s) received no specific funding for this work.

**Competing interests:** The authors have declared that no competing interests exist.

## Conclusion

The NDI and NRS-NP percentage change scores were the most accurate PROMs to measure clinical improvement after surgery for degenerative cervical myelopathy. We recommend using different cut-off estimates for anterior and posterior approach procedures. A MCID achievement rate of 60% or less must be interpreted in the perspective that the main goal of surgery for degenerative cervical myelopathy is to prevent worsening of the condition.

## Introduction

Degenerative cervical myelopathy (DCM) describes a range of conditions in the cervical spine causing cord compression and neurological dysfunction [1]. There is current lack of evidence for nonoperative management in terms of preventing neurological deterioration, although physical rehabilitation and close observation can be considered in mild to asymptomatic cases. For moderate to severe cases, individualized surgical treatment is recommended [2–4]. Anterior Cervical Discectomy and Fusion (ACDF) and Anterior Cervical Disc Arthroplasty (ACDA) are frequently used in patients with disc herniation, while posterior approach procedures are well-established treatments options for patients with posterior and/or multi-level spinal cord compression [5]. In cases where symptoms are caused by spinal cord compression due to cervical ossification of the posterior longitudinal ligament, no treatment consensus is obtained and various anterior and posterior approach procedures are currently applied [6, 7].

The aim of surgery has traditionally been to stop progression of spinal cord dysfunction symptoms. However, recent studies have shown that many patients report improvement post intervention both regarding functionality and disability, as well as quality-of-life outcomes [2, 8]. Depending on PROMs used, severity of preoperative disease and length of follow-up, improvement rates range from around 20 to 80% [9, 10].

Patient-Reported Outcome Measures (PROMs) are commonly used to measure clinical improvement or worsening in spine literature. In combination with the concept of Minimal Clinically Important Difference (MCID), defined as the smallest change in an outcome score that is clinically beneficial within a patient group [11], optimal cut-off estimates for an individual PROM can be assessed [12, 13]. The traditional method is to assess the MCID change score, or the delta value. However, since the interpretation of a change score is dependent on the baseline score, the percentage change score can provide a more representative result at group level [14]. To date, MCID estimates for PROM percentage change scores have not been reported for DCM patients undergoing surgery. Further, there is current lack of evidence in terms of which PROMs are the more accurate in capturing changes in health status among these patients and whether results differ across surgical approach.

The purpose of this study was to estimate MCID for frequently used PROMs 3 and 12 months after surgery for DCM; NDI, Numeric Rating Scale for arm pain (NRS-AP) and neck pain (NRS-NP), Euro-Qol (EQ-5D-3L), and European Myelopathy Score (EMS). A secondary aim was to report achievement rates of MCID through 12 months of follow-up. The MCID estimates are reported for change scores and percentage changes scores for the whole sample, as well as for anterior and posterior approach procedural groups.

## Materials and methods

### Data collection

All data were collected through the Norwegian Registry for Spine Surgery (NORspine) which is a government funded comprehensive clinical registry. Participation in NORspine is not required for a patient to gain access to the health care, or for payment/reimbursement to a provider. All Norwegian health care providers offering cervical spine surgery (six public hospitals and three private clinics) report to NORspine. The proportion of operated patients reported to the registry was 75–78% over the study period [15].

Our research protocol was approved by the Norwegian Committee for Medical and Health Research Ethics Midt (2014/344). Informed consent was obtained from all patients before entering the registry.

### Design

This is a multicenter observational study with follow-up at 3 and 12 months. Results are reported consistent with the Strengthening The Reporting of Observational Studies in Epidemiology (STROBE) statement [16], and methods are in accordance with the COnsensus-based Standards for the selection of health Measurement INstruments (COSMIN) recommendations [12].

### Eligibility criteria

A cohort of 614 patients undergoing surgery for DCM between January 2011 and August 2016 were found to be eligible (Fig 1). Exclusion criteria were: 1) prior surgery the index level; and 2) patients undergoing combined anterior and posterior approach, since these patients commonly are selected on a case-by-case basis [17]. Of the 614 patients, 371 underwent either ACDF (363, 98%) or ACDA (8, 2%), and 243 patients underwent posterior approach procedures, such as laminectomy with or without fusion, hemilaminectomy or laminoplasty.

### Measurements

At admission for surgery (baseline), patients complete the NORspine questionnaire which cover demographics, location and extent of pain and PROMs. During the hospital stay, the surgeon records data concerning diagnosis, American Society of Anesthesiologists physical status (ASA), surgical treatment and comorbidity on a separate form. Under 'indication for operation' the surgeon can checkmark if he/she considers the patient to have myelopathy based on clinical assessment and radiological findings. To avoid selective reporting, the 3- and 12-month follow-up is conducted by the NORspine central registry unit without involvement from treating hospitals. After surgery, a questionnaire identical to that used at baseline is distributed by mail to every registered patient. One reminder questionnaire is sent to those who do not respond. The following PROMs are collected:

1. Neck Disability Index (NDI): a patient-completed questionnaire focusing on the patient's functional status and scores ranging from 0 (no disability) to 100 (greatest disability) [18].

2. Numeric Rating Scale for arm (NRS-AP) and neck pain (NRS-NP): a scale that assesses pain level ranging from 0 (no pain) to 10 (worst conceivable pain) [19].

3. EuroQoL (EQ-5D-3L): a generic measure assessing health-related quality of life with scores ranging from -0.59 (worse than death) to 1 (perfect health) [20].

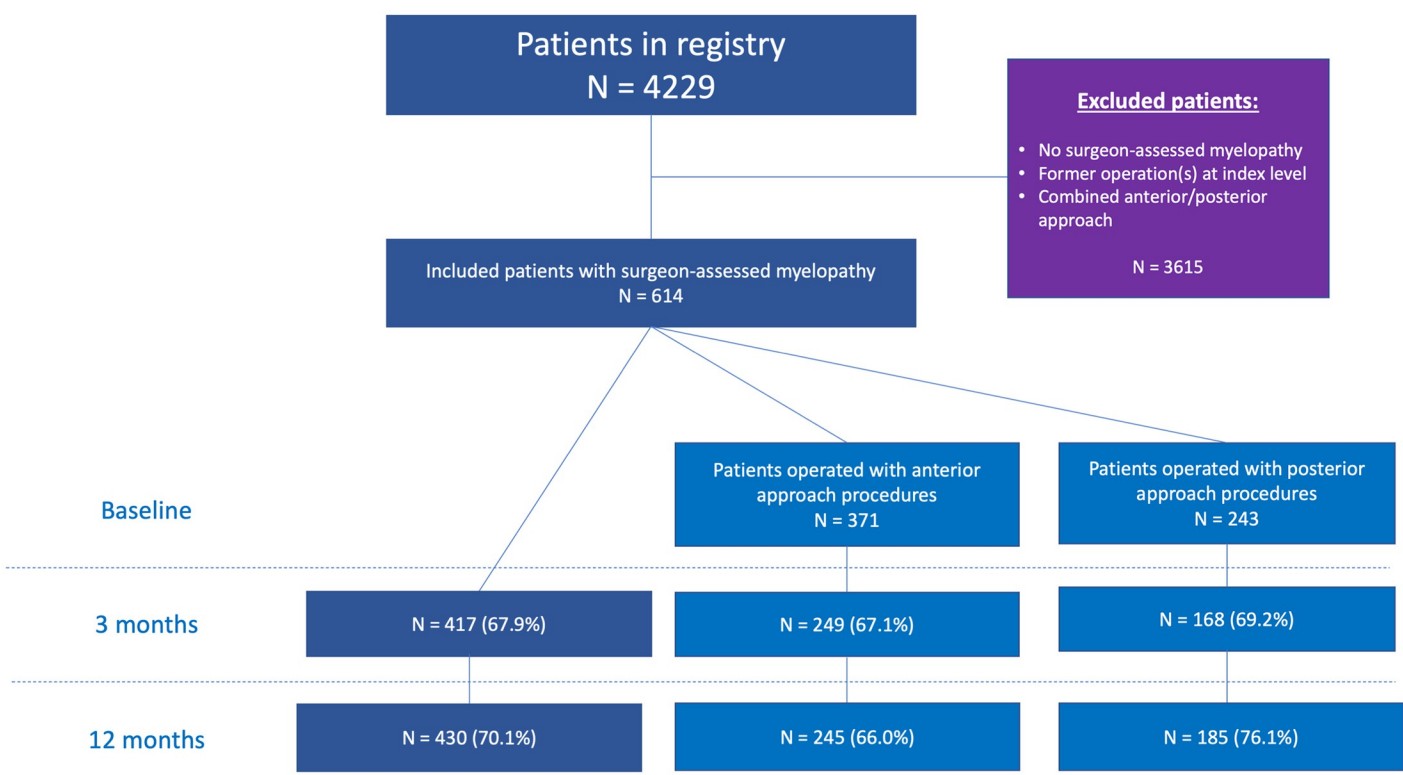

**Fig 1. Project flow chart.** Exclusion criteria for patients included in the study with follow-up rates.

4. European Myelopathy Score (EMS): a patient-based questionnaire derived for assessing spinal cord function. Scoring is between 5 (severe deficit) and 18 (no symptoms) [21].

The Global Perceived Effect scale (GPE) was in the present study used as an external criterion for defining MCID. The GPE measures patient-reported treatment outcome through one single question and seven answer choices; "completely recovered", "much improved", "slightly improved", "unchanged, "slightly worse", "much worse" and "worse than ever" [22]. Patients reporting to be "completely recovered", "much improved" or "slightly improved" (1–3) were classified as having achieved a MCID. Those who considered themselves to be "unchanged" or worse (4–7) were classified as not improved.

## Statistics

All statistical analyses were performed with the Statistical Package for the Social Sciences (SPSS, version 26). Continuous variables were reported as means and standard deviations and categorical variables as numbers and percentages. Differences were evaluated by Chi-square test for categorical variables and by t-tests for continuous variables. PROM change scores were obtained by subtracting the follow-up score from the baseline score. The percentage change score was calculated by dividing the change score with the baseline score and multiplying by 100. To be able to calculate the EQ-5D-3L percentage change score we converted the value range from -0.6 to 1.0 into a relative score from 0 to 100.

The correlations between the GPE scale and the different PROMs were analyzed using the Spearman correlation coefficient. Receiver Operating Curves (ROCs) were used to assess

discriminative ability of each PROM and to define the optimal cut-off with the highest sensitivity and specificity. ROCs were made by plotting the sensitivity against (1 –specificity) for each possible MCID cut-off estimate. The sensitivity refers to the probability of correctly classifying an individual replying "slightly improved" or better (1–3) according to the PROM score. Correspondingly, the specificity refers to the probability of correctly classifying a patient reporting to be "unchanged" or worse as having "not improved" after surgery (4–7). The area under the ROC (AUC) with 95% confidence interval (CI) describes the test's accuracy of correctly classifying a case according to the anchor. The AUC is classified as "acceptable" from 0.7 to 0.8, "excellent" from 0.8 to 0.9 and "outstanding" from 0.9 to 1.0 [23]. To determine MCID cut-off estimates with highest sensitivity and specificity, the closest point to the upper left corner of the ROC-curve was calculated from the coordinates of the curve. Cut-off estimates were assessed for the whole DCM group and for both procedural groups. Lastly, proportions of patients achieving MCID for the whole group and both procedural groups were calculated using the cut-off estimates for each PROM.

## Results

### Respondents and baseline characteristics

Of 4229 consecutive patients undergoing surgery for degenerative disorders in the cervical spine between January 2011 and August 2016, 614 patients were included. Of these patients, 371 underwent an anterior approach procedure, while 243 underwent a posterior approach procedure. A total of 67.9% and 70.1% of patients responded to the 3- and 12-month follow-up questionnaire, respectively (Fig 1). The non-responding patients were slightly younger (p<0.001), less likely to be retired (p<0.001), and more likely to smoke (p<0.001) (Table 1). There were no statistically significant differences in PROM scores, except for the EQ-5D-3L

**Table 1. Baseline characteristics of respondents and non-respondents to 12-month follow-up.**

|  | Respondents N = 430 | | Non-respondents N = 184 | | Sig. (2-tailed)/ chi-square |
|---|---|---|---|---|---|
|  | N |  | N |  |  |
| Age (years); Mean (SD) | 430 | 59.1 (11.9) | 184 | 53.5 (12.2) | <0.001 |
| Female, no (%) | 430 | 167 (38.8) | 184 | 66 (35.9) | 0.488 |
| ASA level (1–4); Mean (SD) | 430 | 2.4 (1.7) | 184 | 2.3 (1.5) | 0.414 |
| Body Mass Index; Mean (SD) | 417 | 27.0 (4.5) | 179 | 27.5 (5.2) | 0.220 |
| Smokers, no (%) | 428 | 106 (24.8) | 184 | 84 (45.7) | <0.001 |
| University/College education | 402 | 137 (31.6) | 173 | 56 (32.4) | 0.823 |
| Retired, no (%) | 430 | 121 (28.1) | 183 | 23 (12.6) | <0.001 |
| Comorbidity, no (%) | 422 | 227 (53.8) | 183 | 109 (59.6) | 0.189 |
| Levels operated, Mean (SD) | 418 | 1.9 (1.1) | 184 | 1.85 (1.1) | 0.376 |
| NDI; Mean (SD) | 428 | 33.7 (17.3) | 178 | 36.6 (17.4) | 0.060 |
| NRS-AP; Mean (SD) | 399 | 5.0 (2.9) | 164 | 5.1 (3.0) | 0.794 |
| NRS-NP; Mean (SD) | 401 | 4.7 (3.0) | 162 | 5.1 (2.9) | 0.134 |
| EQ-5D-3L; Mean (SD) | 392 | 0.47 (0.32) | 171 | 0.39 (0.34) | 0.008 |
| EMS; Mean (SD) | 384 | 14.5 (2.3) | 165 | 14.4 (2.5) | 0.750 |

SD, Standard Deviation; NDI, Neck Disability Index (0–100); NRS-AP Numeric Rating Scale for arm pain (0–10); NRS-NP, Numeric Rating Scale for neck pain (0–10); EQ-5D-3L, Health-Related Quality-of-Life by EuroQol (-0.4– 1.0); EMS, European Myelopathy Score (5–18).

**Table 2. Baseline characteristics of the whole myelopathy group and of the two procedural groups.**

| | Whole myelopathy group N = 430 | | Anterior approach group N = 245 | | Posterior approach group N = 185 | | Sig. (2-tailed)/ chi-square |
|---|---|---|---|---|---|---|---|
| | N | | N | | N | | |
| Age (years); Mean (SD) | 430 | 59.1 (11.9) | 245 | 53.7 (11.0) | 185 | 66.1 (8.9) | <0.001 |
| Female; no (%) | 430 | 167 (38.8) | 245 | 108 (44.1) | 185 | 59 (31.9) | 0.01 |
| ASA level (1–4); Mean (SD) | 430 | 2.4 (1.7) | 245 | 2.0 (1.4) | 185 | 2.9 (1.9) | <0.001 |
| Body Mass Index; Mean (SD) | 417 | 27.0 (5.0) | 363 | 27.3 (4.4) | 178 | 26.8 (5.1) | 0.260 |
| Smokers; no (%) | 425 | 106 (24.9) | 243 | 62 (25.5) | 182 | 44 (24.2) | 0.752 |
| No of levels operated; Mean (SD) | 418 | 1.9 (1.1) | 241 | 1.4 (0.6) | 177 | 2.7 (1.2) | <0.001 |
| Comorbidity; no (%) | 422 | 227 (53.8) | 238 | 110 (46.2) | 184 | 117 (63.6) | <0.001 |
| Currently working; no (%) | 430 | 110 (25.9) | 240 | 85 (35.4) | 184 | 25 (13.6) | <0.001 |
| Retired; no (%) | 430 | 121 (28.1) | 245 | 34 (13.9) | 185 | 87 (47.0) | <0.001 |
| NDI; Mean (SD) | 428 | 33.7 (17.3) | 244 | 33.9 (16.9) | 184 | 33.4 (18.0) | 0.753 |
| NRS-AP; Mean (SD) | 399 | 5.0 (2.9) | 232 | 5.1 (3.0) | 167 | 4.9 (2.9) | 0.442 |
| NRS-NP; Mean (SD) | 401 | 4.7 (3.0) | 234 | 4.9 (2.9) | 167 | 4.4 (3.1) | 0.062 |
| EQ-5D-3L; Mean (SD) | 392 | 0.47 (0.32) | 225 | 0.49 (0.30) | 167 | 0.44 (0.33) | 0.084 |
| EMS; Mean (SD) | 427 | 14.5 (2.4) | 243 | 14.9 (2.2) | 184 | 13.9 (2.5) | <0.001 |

SD, Standard Deviation; NDI, Neck Disability Index (0–100); NRS-AP Numeric Rating Scale for arm pain (0–10); NRS-NP, Numeric Rating Scale for neck pain (0–10); EQ-5D-3L, Health-Related Quality-of-Life by EuroQol (-0.4–1.0); EMS, European Myelopathy Score (5–18).

mean, which was lower (poorer health-related quality-of-life) among non-responders (p<0.008) (Table 1).

Baseline characteristics of the whole myelopathy group and the two procedural groups are presented in Table 2. Compared to the anterior approach procedure group, patients in the posterior approach group were more likely to be male, not working, and to be operated at a higher number of levels. Also, they had significantly higher mean age, higher mean ASA level, more comorbidity, and more severe myelopathy symptoms according to EMS.

## Correlation between the PROMs and the external criterion

For all PROMs, there was a stepwise decrease in mean change scores and mean percentage change scores at 12 months for patients who reported themselves to be completely recovered, much better and slightly better compared to those reporting no change or some degree of worsening (S1 Table). A similar pattern was found for results at 3 months (obtained on request). For the whole group, the Spearman correlation coefficients ranged from 0.30 to 0.59. The NDI showed the strongest correlation with the external anchor.

## AUC and MCID

We found minor differences in AUC and MCID cut-off estimates at 3 and 12 months. Therefore, further analysis of the data is presented only for the PROMs at 12-month follow-up. 3-month scores are presented in S2 Table.

The change scores of NDI, NRS-NP and the EQ-5D-3L showed acceptable AUC values (>0.70), whereas AUC values of the NRS-AP change score and EMS percentage change score were slightly lower than acceptable (0.69 and 0.68, respectively) (Table 3). Most of the AUC change score values (0.64–0.74) were similar to or lower than the corresponding AUC percentage change score value (0.68–0.77). Only for EMS, the change score AUC (0.69) was higher

**Table 3. Area under the curve and cut-off estimates for Minimal Clinically Important Difference for all Patient-Reported Outcome Measures at 12 months.**

| | | Change score (points) | Percentage change score (%) |
|---|---|---|---|
| **NDI** | AUC (95% CI) | 0.74 (0.69, 0.79) | 0.77 (0.72, 0.81) |
| | Cut-off (% sensitivity, % specificity) | 4.3 (0.68, 0.68) | 15.7 (0.71, 0.71) |
| **NRS-AP** | AUC (95% CI) | 0.64 (0.58, 0.70) | 0.69 (0.63, 0.75) |
| | Cut-off (% sensitivity, % specificity) | 0.5 (0.66, 0.53) | 23.6 (0.63, 0.61) |
| **NRS-NP** | AUC (95% CI) | 0.73 (0.67, 0.78) | 0.76 (0.70, 0.81) |
| | Cut-off (% sensitivity, % specificity) | 0.5 (0.71, 0.64) | 15.5 (0.72, 0.71) |
| **EQ-5D-3L** | AUC (95% CI) | 0.70 (0.64, 0.77) | 0.70 (0.64, 0.77) |
| | Cut-off (% sensitivity, % specificity) | 0.02 (0.70, 0.66) | 2.2 (0.68, 0.66) |
| **EMS** | AUC (95% CI) | 0.69 (0.63, 0.75) | 0.68 (0.61, 0.74) |
| | Cut-off (% sensitivity, % specificity) | 0.5 (0.58, 0.69) | 4.2 (0.58, 0.69) |

NDI, Neck Disability Index (0–100); AUC, Area Under the Curve, NRS-AP, Numeric Rating Scale for arm pain (0–10); NRS-NP, Numeric Rating Scale for neck pain (0–10); EQ-5D-3L, Health-Related Quality-of-Life by EuroQol (-0.4–1.0); EMS, European Myelopathy Score (5–18).

than the percentage change score AUC (0.68) (Table 3). The percentage change scores of the NDI and NRS-NP had the highest sensitivity and specificity.

Similar results were found for AUCs analyzed for the anterior and posterior approach groups. However, there was a tendency to lower discriminative ability for all PROMs in the posterior approach group except for EMS in which case the AUCs were higher in this group (Table 4).

## Proportions of patients with clinical improvement at 12-month follow-up

In Fig 2, we present the proportions of patients that achieved a clinical improvement according to MCID estimates for percentage change scores at 12-month follow-up. Overall, NDI (59%),

**Table 4. Minimal Clinically Important Difference cut-off estimates for all Patient-Reported Outcome Measures in the two procedural subgroups at 12 months.**

| | | Anterior approach (% sensitivity, % specificity) | AUC (95% Confidence Interval) | Posterior approach (% sensitivity, % specificity) | AUC (95% Confidence Interval) |
|---|---|---|---|---|---|
| **NDI** | Change score (points) | 5.9 (0.70, 0.70) | 0.74 (0.67, 0.81) | 2.4 (0.68, 0.68) | 0.73 (0.66, 0.81) |
| | Percentage change score (%) | 16.2 (0.72, 0.71) | 0.77 (0.71, 0.84) | 14.4 (0.71, 0.71) | 0.76 (0.68, 0.83) |
| **NRS-AP** | Change score (points) | 0.5 (0.66, 0.52) | 0.66 (0.58, 0.74) | 0.5 (0.65, 0.54) | 0.62 (0.52, 0.72) |
| | Percentage change score (%) | 23.6 (0.64, 0.59) | 0.69 (0.62, 0.77) | 21.1 (0.62, 0.61) | 0.69 (0.60, 0.77) |
| **NRS-NP** | Change score (points) | 0.5 (0.76, 0.62) | 0.77 (0.69, 0.84) | 0.5 (0.63, 0.66) | 0.66 (0.58, 0.75) |
| | Percentage change score (%) | 18.3 (0.73, 0.73) | 0.77 (0.69, 0.85) | 11.8 (0.69, 0.69) | 0.73 (0.65, 0.81) |
| **EQ-5D-3L** | Change score (points) | 0.02 (0.72, 0.71) | 0.74 (0.66, 0.82) | 0.02 (0.67, 0.61) | 0.66 (0.57, 0.76) |
| | Percentage change score (%) | 2.2 (0.70, 0.71) | 0.74 (0.66, 0.82) | 2.3 (0.63, 0.61) | 0.66 (0.57, 0.76) |
| **EMS** | Change score (points) | 0.5 (0.58, 0.66) | 0.67 (0.58, 0.76) | 0.5 (0.59, 0.72) | 0.72 (0.63, 0.80) |
| | Percentage change score (%) | 4.2 (0.58, 0.66) | 0.65 (0.55, 0.74) | 4.2 (0.59, 0.72) | 0.71 (0.62, 0.81) |

AUC, Area Under the Curve; NDI, Neck Disability Index (0–100); NRS-AP, Numeric Rating Scale for arm pain (0–10); NRS-NP, Numeric Rating Scale for neck pain (0–10); EQ-5D-3L, Health-Related Quality-of-Life by EuroQol (-0.4–1.0); EMS, European Myelopathy Score (5–18).

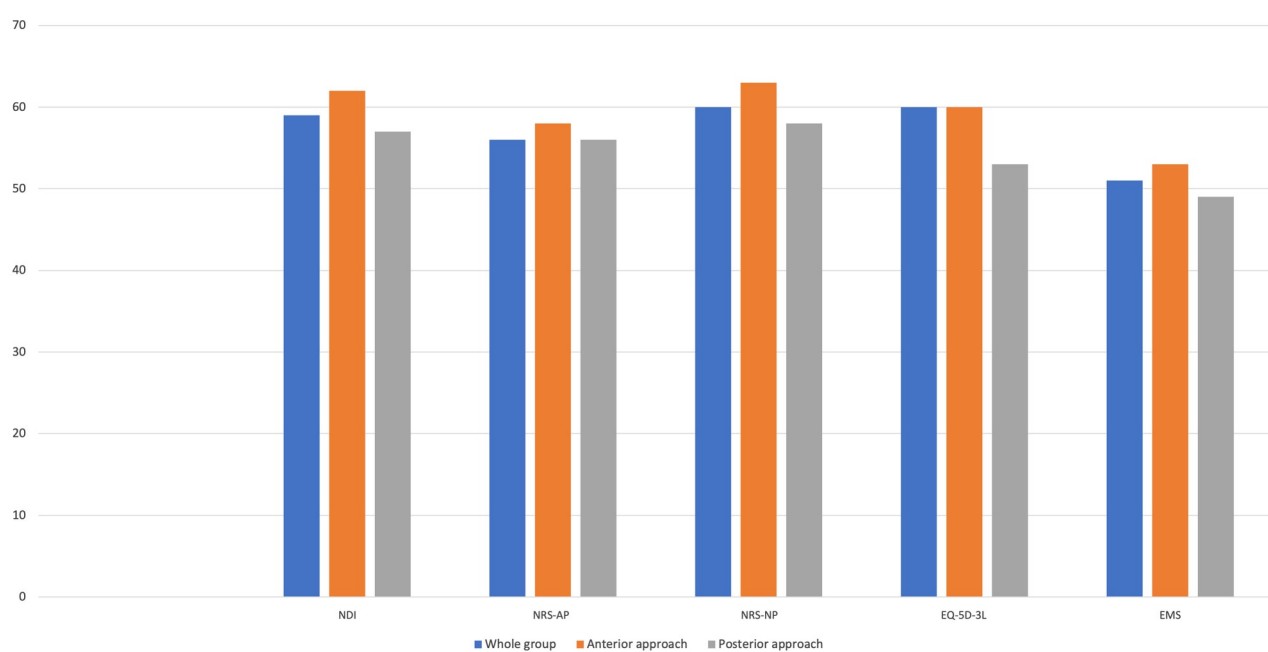

**Fig 2. Clinical improvement rates.** Percentage of patients achieving improvement larger than the Minimal Clinically Important Difference (MCID) according to the Neck Disability Index (NDI), Numeric Rating Scale for arm pain (NRS-AP) and neck pain (NRS-NP), Euro-Quol-5D-3L (EQ-5D-3L) and European Myelopathy Score (EMS). Results are provided by the percentage change score from baseline to 12-month follow-up.

NRS-NP (61%) and EQ-5D-3L (59%) showed similar proportions of achieving a MCID, whereas NRS-NP (56%) and, in particular, EMS (51%) showed lower proportions of improvement. The rates were slightly higher for the anterior approach group compared to the posterior approach group for both change score and percentage change score (S3 Table).

## Discussion

This study showed that NDI and NRS-NP were the most accurate PROMs to measure MCID among patients undergoing surgery due to DCM. EQ-5D-3L also showed acceptable accuracy for both change and percentage change score. Further, achievement of clinical improvement according to the optimal MCID estimates of the investigated PROMs ranged from 51% to 62%, depending on type of PROM, type of MCID and surgical approach.

Although there are several studies investigating MCID for DCM patients undergoing surgery [24–28], there are no reports of percentage change scores for this patient group. In our study, the majority of the percentage change scores were more accurate than the change scores. As shown in Table 3, percentage change scores for NDI, NRS-AP and NRS-NP showed higher AUC, including higher sensitivity and specificity, compared to the change scores. For EQ-5D, the AUCs were identical, while the EMS AUC was slightly higher for the change score than for the percentage change score (0.69 vs. 0.68). Since the use of change scores for benchmarking has been criticized for not taking into account the baseline score [29–31], we recommend using percentage change scores in future research.

The observed MCID estimate of 4.3 points for the NDI 12-month change score is similar to a previous study of Kato et al., who found a cut-off estimate of 4.2 in 101 myelopathy patients undergoing cervical laminoplasty [32]. Chien et al. report a slightly higher cut-off of 6 for NDI which might be due to a very small patient sample (n = 45) [26]. Similarly, in a study of 30

DCM patients by Auffinger et al., five statistical methods used for calculation of cut-off estimates showed similar or substantially higher findings for both NDI (4.8–13.4) and NRS-NP (0.36–3.11) [25].

The accuracy of EQ-5D-3L has also been assessed in a previous study. Kato et al. reported a MCID estimate of 0.05 for EQ-5D-3L with an AUC of 0.704 [32], which is in accordance with the results in the present study. Since the accuracy of EQ-5D-3L has been found to be acceptable (>0.70) in both these studies, we recommend further use of this PROM for DCM patients.

Several studies have reported MCID estimates for degenerative neck surgery patients. However, in many of the investigated cohorts there have been a mix of radiculopathy and myelopathy patients [33–35]. We argue that it is necessary to distinguish between myelopathy and radiculopathy patient cohorts considering the smaller amount of expected improvement among DCM patients. For example, Carreon et al., who analyzed a mixed sample of 505 patients, reported higher MCID estimates than our study for both NDI (7.5 vs. 4.3), NRS-AP (2.5 vs. 0.5) and NRS-NP (2.5 vs. 0.5) [34].

As far as we know, no previous study has presented MCID estimates for EMS and NRS-AP in a DCM cohort.

## Surgical approach

We found minor differences in accuracy of NDI and NRS-NP across patients undergoing anterior versus posterior surgical procedures. However, there was a tendency to lower discriminative ability for NDI, the two NRS scores and EQ-5D-3L in the posterior approach group (Table 4). In each group, NDI and NRS-NP showed the best discriminative ability.

The MCID estimates for NDI, NRS-AP and NRS-NP were lower in the posterior approach group compared to the anterior approach group. This may indicate that posterior approach patients, which were older and had multilevel degenerative disease, were satisfied with less improvement compared to the younger and healthier patients in the anterior approach group. These results confirm that it is reasonable to analyze these two surgical groups separately. They also suggest that the interpretation of change and percentage change scores of PROMs should be different across anterior and posterior procedures.

## Proportion of patients achieving MCID

The proportion of DCM patients that achieved MCID varied between 51% and 61% for the percentage change score. This is in line with a previous study by Stull et al. which reported that 40 to 61% achieve MCID in a sample of 53 DCM patients [9]. Although Stull et al. found little or no difference in achievement rates between radiculopathy and myelopathy patients, others have shown that the proportion of patients achieving a MCID is substantially higher among radiculopathy patients. Applying a cut-off estimate of 15 for NDI, two recent studies found NDI success rates of 80–92% for patients undergoing ACDF or ACDA [36, 37].

## Limitations and strengths

GPE is a self-reported scale and not an objective anchor. This represents the main limitation of our study as global scale ratings tend to be influenced by the current health status of the patient [22]. However, no alternative gold standard currently exists, and the GPE is still the most frequently used anchor in scientific literature [38–42].

The main inclusion criterion for all patients was that the operating surgeon had made a checkmark for myelopathy (yes/no) in the post-operative questionnaire under "indication for operation". This response represents a subjective judgement based on patient history, clinical

features, and radiological findings. Since we have no objective reference for evaluating the accuracy of the surgeons' judgment, misclassifications could exist.

The non-respondent rate of approximately 30% is usually regarded as acceptable for a spine registry [43]. As some of the baseline characteristics of the non-respondents have been associated with poorer outcomes [44], this might still be considered a selection bias especially since we are estimating the proportion of patients achieving MCID. However, this should be of less importance when assessing actual cut-off estimates across a wide range of outcomes. Two previous studies found no differences in outcome when comparing respondents and non-respondents at follow-up, though both had slightly lower non-respondent rates [45, 46].

A major strength of this study is the large sample size of surgical patients in daily clinical practice and the high coverage rate [15] indicating a high external validity of our results.

## Conclusion

NDI and NRS-NP were the most accurate PROMs to measure a clinical improvement according to MCID estimates 12 months after surgery for DCM. Also, EQ-5D-3L showed acceptable discriminative ability.

Percentage change scores were more accurate than change scores, hence, we recommend using percentage change cut-off estimates in future studies. The cut-off estimates and MCID achievement rates were also slightly higher for the anterior approach group compared to the posterior approach group indicating that separate cut-off estimates should be used for each surgical approach.

An achievement of a MCID of 60% or less among DCM patients must be interpreted in the perspective that the main goal of surgery is to prevent worsening of the condition.

## Supporting information

**S1 Table. Mean scores with standard deviation of the Patient-Reported Outcome Measures at 12 months for the whole myelopathy group according to the Global Perceived Effect Scale.** Spearman, Spearman's rank correlation coefficient; Neck Disability Index (0–100); SD, Standard Deviation; NRS-AP, Numeric Rating Scale for arm pain (0–10); NRS-NP, Numeric Rating Scale for neck pain (0–10), EQ-5D-3L, Health-Related Quality-of-Life by EuroQol (-0.4–1.0), EMS, European Myelopathy Score (5–18).
(DOCX)

**S2 Table. Area under the curve and cut-off estimates for "Minimal Clinically Important Difference" for all Patient-Reported Outcome Measures at 3 months.** NDI, Neck Disability Index (0–100); AUC, Area Under the Curve, NRS-AP, Numeric Rating Scale for arm pain (0–10), NRS-NP, Numeric Rating Scale for neck pain (0–10), EQ-5D-3L, Health-Related Quality-of-Life by EuroQol (-0.4–1.0); EMS, European Myelopathy Score (5–18).
(DOCX)

**S3 Table. Proportion of patients with an improvement larger than "Minimal Clinically Important Difference" at 12-months follow-up according to Patient-Reported Outcome Measures.** NDI, Neck Disability Index (0–100), NRS-AP, Numeric Rating Scale for arm pain (0–10), NRS-NP, Numeric Rating Scale for neck pain (0–10), EQ-5D-3L, Health-Related Quality-of-Life by EuroQol (-0.4–1.0), EMS, European Myelopathy Score (5–18).
(DOCX)

## Acknowledgments

The authors would like thank Andrew Malcolm Garratt at the Norwegian Institute of Public Health for helping with the calculation of the EQ-5D-3L percentage change score.

## Author Contributions

**Conceptualization:** Christer Mjåset, John-Anker Zwart, Frode Kolstad, Tore Solberg, Margreth Grotle.

**Data curation:** Christer Mjåset.

**Formal analysis:** Margreth Grotle.

**Methodology:** Tore Solberg.

**Project administration:** Christer Mjåset.

**Supervision:** John-Anker Zwart, Frode Kolstad, Margreth Grotle.

**Validation:** John-Anker Zwart, Frode Kolstad, Tore Solberg.

**Visualization:** Christer Mjåset.

**Writing – original draft:** Christer Mjåset.

**Writing – review & editing:** Christer Mjåset, John-Anker Zwart, Frode Kolstad, Tore Solberg, Margreth Grotle.

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
