## [Decision Letter · Decision Letter 0]

28 Sep 2021

PONE-D-21-25531Clinical Improvement after Surgery for Degenerative Cervical Myelopathy; a Comparison of Patient-Reported Outcome Measures during 12-Month Follow-UpPLOS ONE

Dear Dr. Mjåset,

Thank you for submitting your manuscript to PLOS ONE. After careful consideration, we feel that it has merit but does not fully meet PLOS ONE’s publication criteria as it currently stands. Therefore, we invite you to submit a revised version of the manuscript that addresses the points raised during the review process.

We look forward to receiving your revised manuscript.

Kind regards,

Michael G. Fehlings

Academic Editor

PLOS ONE

Reviewers' comments:

Reviewer's Responses to Questions

**Comments to the Author**

1. Is the manuscript technically sound, and do the data support the conclusions?

Reviewer #1: No

Reviewer #2: Yes

2. Has the statistical analysis been performed appropriately and rigorously? 

Reviewer #1: No

Reviewer #2: No

3. Have the authors made all data underlying the findings in their manuscript fully available?

Reviewer #1: No

Reviewer #2: Yes

4. Is the manuscript presented in an intelligible fashion and written in standard English?

Reviewer #1: Yes

Reviewer #2: Yes

5. Review Comments to the Author

Reviewer #1: The authors present a retrospective review of prospectively-collected data in an attempt to determine the ability of the listed patient reported outcomes to produce a significant clinical improvement (termed “Minimal Important Change”) after surgery for degenerative cervical myelopathy (DCM). These results are based on national registry data, and the outcomes of NDI, NRS for arm and neck, the EQ-5D and the European Myelopathy Score. The data collection methods are clearly presented, and the response to follow up at 12 months is acceptable (70%).

Unfortunately, this article suffers from a number of major flaws, including (but not limited to) the interpretation of the currently available literature on the topic, the selected methodology, the inclusion criteria, as well as the interpretation and representation of the results. These are explained in detail below, however due to the combination of these flaws and lack of novelty, I am unable to support publication of this article.

1 – Introduction – I agree DCM is a progressive disorder, however surgery should not be presented as the ‘standard treatment’. The surgical treatment recommendations varies depending on the severity of myelopathy, judged on a functional outcome assessment such as the Modified Japanese Orthopedic Association score (MJOA). This is a recommendation from the international clinical practice guidelines published in 2017 (Fehlings et al). The references 4 & 5 presented in the article refer to a mixture of quality of life as well as functional outcome measures after surgery specifically after ACDF surgery, in small numbers – there are more appropriate articles to cite regarding outcomes after surgery for DCM (i.e. the CSM-North America and CSM-International studies).

2- Throughout the article the authors refer to the “Minimal Important Change” which I suggest is a misnomer for “Minimum Clinically Important Difference” or “MCID”. The MCID is often cited in literature when discussing patient reported outcome measures, particularly with DCM.

3- The authors state repeatedly that the MCID has not been reported for DCM patients – when there are multiple works that have explored this exact fact- Tetreault et al 2015 for the MJOA, Badhiwala et al 2018 for the SF-36 to name a couple.

4- The authors refer to ‘percentage change score’ and present this as a novel finding, when in fact the ‘Delta’ value (change from baseline) is a more common modality when discussing PROMs in spine surgery and should therefore be used.

5- The authors excluded combined anterior/posterior surgery – with no explanation

6- The authors present a table of demographics for patients initially enrolled (n=614) but not for the patients who were actually followed up at 12 months (n=430).

7- The authors report a high number of Tobacco smokers in both anterior and posterior groups – and at no point is this fact addressed, adjusted for, or even commented on when it comes to the results and discussion.

8- It is not stated the proportion of patients undergoing disc arthroplasty vs ACDF – I would argue the use of arthroplasty is not internationally substantiated in the treatment of DCM.

9-The statistical analysis is incorrect – the use of T tests when attempting to report the potential outcomes of PROMs standardized to a binary outcome is not going to produce reliable results.

Odds ratios and, at the very least, linear regression modelling is the best initial modality to interrogate the data and represents the most suitable first approach. Multiple regression analysis should then be employed to adjust for confounders such as smoking, gender, number of operated levels, age – all of which have been consistently demonstrated to affect the outcomes after surgery for DCM.

All of this should be performed and validated before any attempt at AUC analysis.

10- Given the above – none of the AUC analysis is accurate.

11- No attempt was made to internally validate the data presented.

12- The authors state the sample size (417) is large. ROC and AUC analysis require significantly larger sample sizes to produce reliable results. See this recent article:

Wilson et al 2020 Frailty Is a Better Predictor than Age of Mortality and Perioperative Complications after Surgery for Degenerative Cervical Myelopathy: An Analysis of 41,369 Patients from the NSQIP Database 2010–2018

Reviewer #2: Page 3 Line 4 Clarify this sentence please- surgery is recommended in moderate-severe at the moment by the AO, however evidence suggests that mild cases benefit too.

Line 9: Comment on OPLL please

Line 12: What incidence of patient report an improvement.

Line 17: Its not more correct, perhaps could be more representative or easier to interpret.

Page 4 line 2: Why was mJOA not evaluated?

Line 9: what was the distribution ACDF/ ACDA?

Page 7 Line 1 How did the authors check for normality?

Page 15 Line 5: More accurate needs to be qualified more comprehensively please

Page 16 Line 15: How do you discriminate between the effects of ant/post approach and age? Summarise here please

6. PLOS authors have the option to publish the peer review history of their article (what does this mean?). If published, this will include your full peer review and any attached files.

Reviewer #1: **Yes: **Jamie R F Wilson

Reviewer #2: No

---

## [Author Response · Author response to Decision Letter 0]

4 Dec 2021

Reviewer #1: 

General comment:

The authors present a retrospective review of prospectively-collected data in an attempt to determine the ability of the listed patient reported outcomes to produce a significant clinical improvement (termed “Minimal Important Change”) after surgery for degenerative cervical myelopathy (DCM). These results are based on national registry data, and the outcomes of NDI, NRS for arm and neck, the EQ-5D and the European Myelopathy Score. The data collection methods are clearly presented, and the response to follow up at 12 months is acceptable (70%).

Unfortunately, this article suffers from a number of major flaws, including (but not limited to) the interpretation of the currently available literature on the topic, the selected methodology, the inclusion criteria, as well as the interpretation and representation of the results. These are explained in detail below, however due to the combination of these flaws and lack of novelty, I am unable to support publication of this article.

Answer to general comment:

In our article, we have found cut-off estimates for PROMs for cervical degenerative myelopathy patients undergoing surgery. This is not an attempt to produce a predictive model for DCM patients. Although we believe such a model is needed, this is beyond the scope of this article.

Since reviewer #1 bring up multiple discussions in spine literature, we have chosen to reference our answers and added a reference list at the very end of the document. 

Comment #1:

Introduction – I agree DCM is a progressive disorder, however surgery should not be presented as the ‘standard treatment’. The surgical treatment recommendations vary depending on the severity of myelopathy, judged on a functional outcome assessment such as the Modified Japanese Orthopedic Association score (MJOA). This is a recommendation from the international clinical practice guidelines published in 2017 (Fehlings et al). The references 4 & 5 presented in the article refer to a mixture of quality of life as well as functional outcome measures after surgery specifically after ACDF surgery, in small numbers – there are more appropriate articles to cite regarding outcomes after surgery for DCM (i.e. the CSM-North America and CSM-International studies). 

Answer: 

We agree that more appropriate references are needed. The articles mentioned have been referenced. The reason for referencing the two original articles was that they both report MCID attainment rates for DCM patients undergoing surgery [1, 2]. Since Reivewer #2 asks for these figures, we have chosen to split the information in two sentences in the revised manuscript (page 3, line 22 – page 4, line 2).

Comment #2: 

Throughout the article the authors refer to the “Minimal Important Change” which I suggest is a misnomer for “Minimum Clinically Important Difference” or “MCID”. The MCID is often cited in literature when discussing patient reported outcome measures, particularly with DCM. 

The term Minimal Important Change/”MIC” is described in depth in the article by Mokkink et al from 2010 and is based on the work made by the COSMIN group (COnsensus-based Standards for the selection of health status Measurement INstruments) that aimed to recommend methodological measurement properties for studies of health-related patient reported outcomes [3]. As stated in this article and by de Vet et al [4], literature often interchanges the terms MIC and MCID. However, it has been proposed that MCID is to be used as a term for cross-sectional between-person differences, while MIC is the minimal change in the scoring measure/PROM for a group of patients that is perceived as beneficial. This has been nicely illustrated in Figure 1 in the 2017 article by Clement et al [5]. 

In spine literature, MCID is the preferred term [6]. Therefore, we have decided to follow the advice of the reviewer and apply the term MCID in our article instead of MIC.

Comment #3 

The authors state repeatedly that the MCID has not been reported for DCM patients – when there are multiple works that have explored this exact fact- Tetreault et al 2015 for the MJOA, Badhiwala et al 2018 for the SF-36 to name a couple. 

Answer:

The authors acknowledge that there are multiple studies reporting MCID for DCM patients, and we would like to challenge the reviewer’s comment: 

• We are referring to several articles that have reported MCID for neck surgery patients in the discussion section (page 16-17 in the revised manuscript). We have particularly focused on the studies reporting cut-off estimates for the PROMs in our study (NDI, arm and neck pain, EQ-5D and EMS). 

• On page 3, line 18 in the original manuscript, we state: “To date, MIC estimates for PROM percentage change scores have not been reported for DCM patients”. Also, on page 15, line 3, we state again: “To our knowledge, the percentage change scores for patients undergoing surgery for DCM have not previously been reported”. We believe this is true. We have found no percentage change scores in literature, and this article is the first to report such scores. We are not claiming that MCID estimates in general have not previously been reported. In fact, we state on page 15, line 19 in the original manuscript: “Several studies have reported MIC estimates for degenerative neck surgery patients”.

To clarify this to the reader, we have added a sentence in the discussion section (page 16, line 3-4 in the revised manuscript) stating that there are multiple studies investigating MCID for DCM patients, but that we find no other study reporting “percentage change scores”.

Comment #4 

The authors refer to ‘percentage change score’ and present this as a novel finding, when in fact the ‘Delta’ value (change from baseline) is a more common modality when discussing PROMs in spine surgery and should therefore be used.

Answer:

We agree that the change score or the delta value is a more common modality. In fact, in our article, we are reporting both the change score/the delta value, as well as the percentage change score for each PROM. Since the percentage change score take into account the baseline score, we find that the results are more accurate. This is in line with a previous study where we investigated cut-off scores for a radiculopathy patient group [7]. Similar findings have also been reported for lumbar spine patients [8, 9]. In the discussion section, we, therefore, argue that the percentage change score should be more frequently used in the future (page 16, line 9-11 in the revised manuscript).

Comment #5

The authors excluded combined anterior/posterior surgery – with no explanation

Answer:

We agree that the exclusion of the anterior-posterior surgery patients needs an explanation. 

A combined anterior-posterior approach is not a common treatment of DCM. According to Kim et al [10], patients should be selected for this kind of procedure on a case-by-case basis. Indications include acute spinal trauma, post-laminectomy kyphosis, kyphotic deformity with intact posterior tension band, multilevel spondylosis and OPLL, and pre-existing risk factors for pseudarthrosis. We believe this cohort is substantially different from the general myelopathy cohort and should be investigated separately. 

A comment regarding this fact has been added to the article on page 6, line 5-6.

Comment #6:

The authors present a table of demographics for patients initially enrolled (n=614) but not for the patients who were actually followed up at 12 months (n=430). 

Answer:

We agree that this information will be useful and have made edits to the article accordingly. Please see edits in the manuscript on page 10, line 9, and in Table 2 on page 10-11.

Comment #7:

The authors report a high number of Tobacco smokers in both anterior and posterior groups – and at no point is this fact addressed, adjusted for, or even commented on when it comes to the results and discussion. 

Answer:

Again, as mentioned in the introduction, we are not attempting to develop a prognostic model for DCM patients in this study. Therefore, we believe that the characteristics of the patients in the cohort is of less importance than the actual results. Also, a comparable study by Fehlings et al. report similar findings regarding tobacco use in their cohort (27.35% vs. 24.9%) [11]. Therefore, we have not made any edits in the article related to this comment..

Comment #8:

It is not stated the proportion of patients undergoing disc arthroplasty vs ACDF – I would argue the use of arthroplasty is not internationally substantiated in the treatment of DCM. 

Answer:

In our study, 8 patients underwent disc arthroplasty. All of them were younger than the mean and had surgery in only one and two levels. The use of arthroplasty has been shown to have similar results as ACDF procedures in several studies of patients with symptoms of radiculopathy or/and myelopathy, e.g., the articles of Gornet et al [12, 13]. We, therefore, believe the inclusion of these few procedures are justified. 

Comment #9: 

The statistical analysis is incorrect – the use of T tests when attempting to report the potential outcomes of PROMs standardized to a binary outcome is not going to produce reliable results. Odds ratios and, at the very least, linear regression modelling is the best initial modality to interrogate the data and represents the most suitable first approach. Multiple regression analysis should then be employed to adjust for confounders such as smoking, gender, number of operated levels, age – all of which have been consistently demonstrated to affect the outcomes after surgery for DCM.

All of this should be performed and validated before any attempt at AUC analysis. 

Answer:

In this study, we are reporting PROM cut-off estimates for patients undergoing surgery for degenerative cervical myelopathy. Also, we have investigated the accuracy of each PROM in the Norwegian Spine Registry for the DCM group. This method is quite standardized. Several studies have performed similar investigations for other types of spine patients undergoing surgery [7, 14, 15]. We have not attempted to do a multiple regression analysis. However, based on our cut-off estimates such an analysis will be possible to do in the future.

Comment #10:

Given the above – none of the AUC analysis is accurate.

Answer:

Based on the argument above, we believe the AUC analysis is accurate. 

Comment #11:

No attempt was made to internally validate the data presented. 

Answer:

In a prediction analysis, validation of the initial result is necessary. However, in our case, we are not attempting to do such an analysis and only trying to produce cut-off estimates for a designated cohort. Our findings need to be validated if one is to use the cut-off estimates for a different cohort. However, that is beyond the scope of this paper.

Comment #12 

The authors state the sample size (417) is large. ROC and AUC analysis require significantly larger sample sizes to produce reliable results. See this recent article:

Wilson et al 2020 Frailty Is a Better Predictor than Age of Mortality and Perioperative Complications after Surgery for Degenerative Cervical Myelopathy: An Analysis of 41,369 Patients from the NSQIP Database 2010–2018 

Answer:

Wilson et al has done a predictive analysis of a DCM cohort. In the Wilson et al article, the objectives are as follows: «(1) define the effect of age on the perioperative outcomes of mortality, unplanned readmission/reoperation, major complication, length of stay and discharge to non-home destination for patients undergoing surgery for DCM, (2) directly compare measures of frailty in the same cohort to determine which factor exhibits a greater influence on the observed outcomes, and (3) define the potential correlation between MFI-5 and MFI-11 in DCM patients» [16]. We do not attempt to develop a predictive model in our article, only produce cut-off estimates for patients.

Also, other recent studies investigating cut-off estimates for similar DCM cohorts and reviewed in the discussion section on page 16-17 (in the revised manuscript), have similar or smaller sample sizes than ours. We therefore believe it is justified to rate our sample size (614) as «large».

Reviewer #2: 

Comment #1: 

Page 3 Line 4 Clarify this sentence please - surgery is recommended in moderate-severe at the moment by the AO, however evidence suggests that mild cases benefit too.

Answer:

We agree that this sentence needs clarifications and have made edits accordingly with proper references in the introduction on page 3, line 10-13.

Comment #2

Line 9: Comment on OPLL please

Answer: 

Information and relevant references have been added regarding the treatment of OPLL in the introduction on page 3, line 17-19.

Comment #3

Line 12: What incidence of patient report an improvement.

Answer: 

We believe what the reviewer is asking for is the MCID attainment rate for degenerative cervical myelopathy patients undergoing surgery. The figures reported by Stull et al. and Goh et al. are now inserted in the text and referenced accordingly (page 4, line 1-2).

Comment #4

Line 17: It’s not more correct, perhaps could be more representative or easier to interpret.

Answer:

We agree that this sentence can be misleading and have changed the wording to: “… the percentage change score can provide a more representative result at group level” (page 4, line 9).

Comment #5

Page 4 line 2: Why was mJOA not evaluated?

Answer:

Although mJOA is a widely used PROM, the use is not frequent in Scandinavia. Also, mJOA is not included in the patient questionnaire of the Norwegain Spine Registry and could followingly not be included in the study. 

Comment #6:

Line 9: what was the distribution ACDF/ ACDA?

Answer:

As mentioned in the aswers to Reviewer #1, the majority of the anterior approach procedures were ACDF (363, 98%) and only 8 (2%) ACDAs were performed. The information has been added to the text on page 6, line 7.

Comment #7:

Page 7 Line 1 How did the authors check for normality?

Answer:

We examined all continuous variables with respect to normality by using histograms and by analyzing skewness and kurtosis. All variables were normally distributed. 

Comment #8:

Page 15 Line 5: More accurate needs to be qualified more comprehensively please

Answer:

If we understand the reviewer correctly, there is need for a quantification of the term “accuracy”.

In general, the percentage change scores were more accurate than the change scores. Table 3 shows that the majority of percentage change scores in general had higher AUC values, including higher sensitivity and specificity values, than change scores of the majority of PROMs.

Please see edits on page 16, line 4-9. If we have misinterpreted the question, please clarify again. 

Comment #9

Page 16 Line 15: How do you discriminate between the effects of ant/post approach and age? Summarize here please.

Answer:

We are not sure how to interpret this comment. However, if the question is whether surgery OR age influence the proportion of improvement according to MCIDs, our study results do not reflect this issue directly. We would need a RCT in order to respond properly to this question. We have, however, conducted a logistic regression analysis in which we investigated to what extent surgery approach, age and the interaction between these two factors predicted the proportion of improvements when using the MCID for NDI (4.30 cut-off value). We found that neither surgical approach (p=0,17), age (p=0,22) or the interaction between these two factors (p=0,26) predicted this outcome. 

If this is not the right interpretation of the reviewer’s question, please specify the comment.

References:

1. Goh GS, Liow MHL, Ling ZM, Soh RCC, Guo CM, Yue WM, Tan SB, Chen JL (2020) Severity of Preoperative Myelopathy Symptoms Affects Patient-reported Outcomes, Satisfaction, and Return to Work After Anterior Cervical Discectomy and Fusion for Degenerative Cervical Myelopathy. Spine (Phila Pa 1976) 45:649-656. doi: 10.1097/brs.0000000000003354

2. Stull JD, Goyal DKC, Mangan JJ, Divi SN, McKenzie JC, Casper DS, Okroj K, Kepler CK, Vaccaro AR, Schroeder GD, Hilibrand AS (2020) The Outcomes of Patients With Neck Pain Following ACDF: A Comparison of Patients With Radiculopathy, Myelopathy, or Mixed Symptomatology. Spine (Phila Pa 1976) 45:1485-1490. doi: 10.1097/brs.0000000000003613

3. Mokkink LB, Terwee CB, Knol DL, Stratford PW, Alonso J, Patrick DL, Bouter LM, de Vet HC (2010) The COSMIN checklist for evaluating the methodological quality of studies on measurement properties: a clarification of its content. BMC Med Res Methodol 10:22. doi: 10.1186/1471-2288-10-22

4. de Vet HC, Ostelo RW, Terwee CB, van der Roer N, Knol DL, Beckerman H, Boers M, Bouter LM (2007) Minimally important change determined by a visual method integrating an anchor-based and a distribution-based approach. Qual Life Res 16:131-142. doi: 10.1007/s11136-006-9109-9

5. Clement ND, Weir D, Holland J, Gerrand C, Deehan DJ (2019) Meaningful changes in the Short Form 12 physical and mental summary scores after total knee arthroplasty. Knee 26:861-868. doi: 10.1016/j.knee.2019.04.018

6. Chung AS, Copay AG, Olmscheid N, Campbell D, Walker JB, Chutkan N (2017) Minimum Clinically Important Difference: Current Trends in the Spine Literature. Spine (Phila Pa 1976) 42:1096-1105. doi: 10.1097/BRS.0000000000001990

7. Mjåset C, Zwart J-A, Goedmakers CMW, Smith TR, Solberg TK, Grotle M Criteria for Success after Surgery for Cervical Radiculopathy; Estimates for a Substantial Amount of Improvement in Core Outcome Measures. The Spine Journal. doi: 10.1016/j.spinee.2020.05.549

8. Werner DAT, Grotle M, Gulati S, Austevoll IM, Lonne G, Nygaard OP, Solberg TK (2017) Criteria for failure and worsening after surgery for lumbar disc herniation: a multicenter observational study based on data from the Norwegian Registry for Spine Surgery. Eur Spine J 26:2650-2659. doi: 10.1007/s00586-017-5185-5

9. Austevoll IM, Gjestad R, Grotle M, Solberg T, Brox JI, Hermansen E, Rekeland F, Indrekvam K, Storheim K, Hellum C (2019) Follow-up score, change score or percentage change score for determining clinical important outcome following surgery? An observational study from the Norwegian registry for Spine surgery evaluating patient reported outcome measures in lumbar spinal stenosis and lumbar degenerative spondylolisthesis. BMC Musculoskelet Disord 20:31. doi: 10.1186/s12891-018-2386-y

10. Kim PK, Alexander JT (2006) Indications for circumferential surgery for cervical spondylotic myelopathy. Spine J 6:299s-307s. doi: 10.1016/j.spinee.2006.04.025

11. Fehlings MG, Ibrahim A, Tetreault L, Albanese V, Alvarado M, Arnold P, Barbagallo G, Bartels R, Bolger C, Defino H, Kale S, Massicotte E, Moraes O, Scerrati M, Tan G, Tanaka M, Toyone T, Yukawa Y, Zhou Q, Zileli M, Kopjar B (2015) A global perspective on the outcomes of surgical decompression in patients with cervical spondylotic myelopathy: results from the prospective multicenter AOSpine international study on 479 patients. Spine (Phila Pa 1976) 40:1322-1328. doi: 10.1097/brs.0000000000000988

12. Gornet MF, Burkus JK, Shaffrey ME, Schranck FW, Copay AG (2019) Cervical disc arthroplasty: 10-year outcomes of the Prestige LP cervical disc at a single level. J Neurosurg Spine:1-9. doi: 10.3171/2019.2.Spine1956

13. Gornet MF, Lanman TH, Burkus JK, Hodges SD, McConnell JR, Dryer RF, Copay AG, Nian H, Harrell FE (2017) Cervical disc arthroplasty with the Prestige LP disc versus anterior cervical discectomy and fusion, at 2 levels: results of a prospective, multicenter randomized controlled clinical trial at 24 months. Journal of Neurosurgery: Spine SPI 26:653-667. doi: 10.3171/2016.10.SPINE16264

14. Hermansen E, Myklebust TA, Austevoll IM, Rekeland F, Solberg T, Storheim K, Grundnes O, Aaen J, Brox JI, Hellum C, Indrekvam K (2019) Clinical outcome after surgery for lumbar spinal stenosis in patients with insignificant lower extremity pain. A prospective cohort study from the Norwegian registry for spine surgery. BMC Musculoskelet Disord 20:36. doi: 10.1186/s12891-019-2407-5

15. Solberg T, Johnsen LG, Nygaard OP, Grotle M (2013) Can we define success criteria for lumbar disc surgery? : estimates for a substantial amount of improvement in core outcome measures. Acta Orthop 84:196-201. doi: 10.3109/17453674.2013.786634

16. Wilson JRF, Badhiwala JH, Moghaddamjou A, Yee A, Wilson JR, Fehlings MG (2020) Frailty Is a Better Predictor than Age of Mortality and Perioperative Complications after Surgery for Degenerative Cervical Myelopathy: An Analysis of 41,369 Patients from the NSQIP Database 2010-2018. J Clin Med 9:3491. doi: 10.3390/jcm9113491

---

## [Decision Letter · Decision Letter 1]

21 Feb 2022

Clinical Improvement after Surgery for Degenerative Cervical Myelopathy; a Comparison of Patient-Reported Outcome Measures during 12-Month Follow-Up

PONE-D-21-25531R1

Dear Dr. Mjåset,

We’re pleased to inform you that your manuscript has been judged scientifically suitable for publication and will be formally accepted for publication once it meets all outstanding technical requirements.

Kind regards,

Michael G. Fehlings

Academic Editor

PLOS ONE

Additional Editor Comments (optional):

Reviewers' comments:

Reviewer's Responses to Questions

**Comments to the Author**

1. If the authors have adequately addressed your comments raised in a previous round of review and you feel that this manuscript is now acceptable for publication, you may indicate that here to bypass the “Comments to the Author” section, enter your conflict of interest statement in the “Confidential to Editor” section, and submit your "Accept" recommendation.

Reviewer #2: All comments have been addressed

2. Is the manuscript technically sound, and do the data support the conclusions?

Reviewer #2: Yes

3. Has the statistical analysis been performed appropriately and rigorously? 

Reviewer #2: Yes

4. Have the authors made all data underlying the findings in their manuscript fully available?

Reviewer #2: Yes

5. Is the manuscript presented in an intelligible fashion and written in standard English?

Reviewer #2: No

6. Review Comments to the Author

Reviewer #2: A much improved manuscript which is clearer in its aims and in its conclusions. The authors defend their manuscript in the response to reviewers. There remains several typographical and grammatical errors, otherwise the report is satisfactory at this stage.

7. PLOS authors have the option to publish the peer review history of their article (what does this mean?). If published, this will include your full peer review and any attached files.

Reviewer #2: No

---

## [Editor Report · Acceptance letter]

28 Feb 2022

PONE-D-21-25531R1 

Clinical Improvement after Surgery for Degenerative Cervical Myelopathy; a Comparison of Patient-Reported Outcome Measures during 12-Month Follow-Up 

Dear Dr. Mjåset:

I'm pleased to inform you that your manuscript has been deemed suitable for publication in PLOS ONE. Congratulations! Your manuscript is now with our production department. 

Kind regards, 

on behalf of

Dr. Michael G. Fehlings 

Academic Editor

PLOS ONE